# The Link between the Demographic and Clinical Factors and Fatigue Symptoms among Rheumatoid Arthritis Patients

**DOI:** 10.3390/ijerph192214681

**Published:** 2022-11-09

**Authors:** Katarzyna Anna Kozłowska, Dorota Formanowicz, Grażyna Bączyk

**Affiliations:** 1Department of Nursing Practices, Poznan University of Medical Sciences, 61-701 Poznan, Poland; 2Department of Medical Chemistry and Laboratory Medicine, Poznan University of Medical Sciences, 61-701 Poznan, Poland

**Keywords:** fatigue, rheumatoid arthritis, clinical factors, demographic factors

## Abstract

Rheumatoid arthritis (RA) is a chronic systemic disease of connective tissue with periods of exacerbation and remission. Fatigue is excessive strain throughout the body that is disproportionate or unrelated to an activity or lifestyle. Fatigue is an integral part of RA in most patients. The study aimed to assess the level of fatigue in RA patients and establish the relationship between fatigue and demographic and clinical factors. The study group consisted of 128 RA patients according to European League Against Rheumatism (EULAR) criteria. The Functional Assessment of Chronic Illness Therapy-Fatigue and -Medical Outcomes Study Short Form 36 (SF-36) vitality scores were used to assess the severity of fatigue symptoms. The analyzed variables were gender, age, disease duration, education, marital status, place of residence, work and residence status, pharmacological treatment, pain, morning stiffness, hemoglobin, C-reactive protein (CRP), rheumatoid factor (RF), compression soreness, Richie Articular Index, and DAS28 disease activity. The examined patients experience chronic fatigue—the mean value on the FACIT-F scale was 24.1 ± 9.1 points and on the SF-36 Vitality score was 14.2 ± 1.8 points. There is a relationship between the level of fatigue and pain, long-lasting morning stiffness, active disease, increased soreness of joints, and low hemoglobin values. When analyzing the symptom of fatigue, each patient should be approached individually, using the existing questionnaires or asking key questions to recognize the situation. The presence of fatigue symptoms should be considered during therapy and patient care by searching for and eliminating additional, intensifying stimuli and increasing its level.

## 1. Introduction

Rheumatoid arthritis (RA) is a chronic systemic disease of connective tissue with an immune background characterized by nonspecific symmetrical arthritis, extra-articular lesions, and systemic complications, leading to disability, disability, and premature death [1]. It affects women more often than men and is most often diagnosed after age 16. The etiology of the disease is still not fully understood. It is characterized by inflammatory changes in the synovial membrane of joints, cartilage, and bones and may also manifest itself extra-articularly. According to research, genetic factors and immune and environmental disorders, often coexisting with each other, influence the development of RA. Pain and swelling in symmetrical joints, along with accompanying joint stiffness, are the predominant symptoms of the disease. Extra-articular changes include, among others, the heart (ischemic heart disease, pericarditis), lungs (pleurisy, bronchiolitis), kidneys (pyelonephritis, amyloidosis), or eyes (scleritis, dry exfoliative keratitis) [1,2,3]. In addition to the symptoms listed above, fatigue occurs in over 70 percent of patients [4]. Although it is a subjective symptom that depends on many factors, it significantly impacts the patient’s well-being and functioning [4]. 

At the international Outcome Measures in Rheumatology (OMERACT) conference in 2012, it was stated that the symptom of fatigue should always be measured in all clinical trials in patients with rheumatoid arthritis to supplement the basic clinical history. Fatigue has been recognized as an integral part of RA in most patients. The priority was also to identify the factors influencing its level [5].

Fatigue is an overwhelming, debilitating, and persistent sense of exhaustion that reduces the ability to carry out daily activities, including working effectively and functioning at the usual level in family or social roles. They can also be perceived as unpleasant, excessive fatigue of the whole body that is disproportionate or unrelated to activity (e.g., work) or effort, lasting more than a month. Sleep and rest cannot reduce this feeling and restore balance to the body. Fatigue affects the mental, physical, cognitive, emotional, and social spheres [6,7,8].

Hewlett et al. [9] created a conceptual model of fatigue-related RA. It is made up of three areas of interaction: The clinical picture of RA disease (e.g., disease activity, treatment, muscle pain, and tenderness), cognitive and behavioral processes (e.g., thoughts—must, should; feelings—anxiety, depression; behavior—hyperactivity or lack of any activity) and personal life (e.g., social roles, job, relationships, the influence of external and internal environmental factors) [9]. Fatigue influences one’s self-efficacy, motivation to act, or mobilization. It may hurt treatment-reluctance or lack of strength to take up physical activity, take medication, and have check-ups with a doctor [6,7,8].

The research aimed to assess the level of fatigue in patients with RA and establish the relationship between fatigue and demographic factors (sex, age, education, marital status, place of residence, professional status, residence status) and clinical factors (duration of the disease, pharmacological treatment (biological/classic), pain intensity, morning stiffness, hemoglobin (Hgb), C-reactive protein (CRP), rheumatoid factor (RF), Ritchie Articular Index compression soreness, and Disease Activity Score in 28 Joints (DAS28).

## 2. Materials and Methods

### 2.1. Study and Participants

The data used for the analysis, as part of a cross-sectional study, were collected in 2016–2018 in the Rheumatology Ward of the M. Dega Orthopedic and Rehabilitation Clinical Hospital of Poznan University of Medical Sciences and the Józef Struś Poznań Multidisciplinary City Hospital.

The study group consisted of 128 patients admitted to the rheumatology departments to monitor the effects of their treatment. Patients diagnosed with RA according to the European League Against Rheumatism (EULAR) in 2010 criteria [10], who had the necessary laboratory test results, and who understood the meaning of the questions asked were eligible for the study. The exclusion criterion was the occurrence of diseases associated with increased fatigue intensity, such as malignancy and mental illness. An additional exclusion criterion was the lack of consent or resignation during the investigation and failure to meet any of the above points. Study participants were asked to fill in the questionnaires with the participation of the researcher or independently. 

### 2.2. Methods/Questionnaires Used

Two standardized questionnaires and an authoring tool were used to achieve the study’s objectives. All respondents completed the same questionnaires. To assess the severity of fatigue symptoms, the Polish versions of the following scales were used: Functional Assessment of Chronic Illness Therapy-Fatigue (FACIT-F) and 36-Item Short Form Survey (SF-36) in the field VITALITY—Medical Outcomes Study Short Form 36 vitality scores (SF-36 v.s.), with the consent of the authors.

The FACIT-F scale (version 4) measures the level of fatigue in various physical, mental, and social areas. It allows for assessing the impact of fatigue on the daily functioning of patients and the related activities of daily living observed during the last week in patients with chronic diseases. The respondent assesses the degree of intensity of a given feeling, marking whether it takes place and, if so, what its power is. The scale consists of 13 questions, scored from 0 to 4. A minimum of 0 points can be obtained at 52 points (the higher the score, the less severe the symptoms of fatigue) [11].

The SF-36 questionnaire is used for subjective quality of life assessment. For the SF-36 scale studies, only the Vitality score domain was used. It consists of 4 questions regarding feelings of vigor, energy, exhaustion, and fatigue. The respondent can choose from 6 answers. A maximum of 24 points can be obtained, and a minimum of 4 points (a small number of points means a low quality of life) [12].

A proprietary questionnaire was used to assess demographic factors, containing questions regarding gender, age, disease duration, current treatment (classic nonsteroidal anti-inflammatory drugs (NSAIDs), disease-modifying drugs, glucocorticosteroids), and biological treatment (yes/no).

DAS28 was used to assess clinical variables and the Visual Analogue Scale (VAS) scale (0–10 cm) was used to determine the severity of joint pain, with a scale used to determine joint tenderness and swelling.

The assessment was performed by a rheumatologist (in the presence of the first co-author of the manuscript, who assisted in this assessment) during the interview and physical examination of the patient. A medical calculator (DAS 28 CRP 3) version 1.1 [13] was used. The tender joint count according to the Ritchie Articular Index and the swollen joint count and CRP value were entered into the calculator.

Moreover, the participants were asked about the duration of the morning stiffness symptom (in minutes) and the hemoglobin concentration (Hgb), CRP, and rheumatoid factor (RF) titer.

### 2.3. Ethics

The study was conducted following the Helsinki Declaration and was approved by the Ethics Committee of the Poznan University of Medical Sciences, registered under reference numbers 46/16 and 564/16. Participation in the survey was voluntary and anonymous. All participants in the study gave their informed consent to participate in it. The informed consent form contained information about the study, its purpose, the method of answering the questions, and the possibility of withdrawing from the study at any time without suffering consequences. 

### 2.4. Statistical Analysis

Qualitative variables are described using the count (*n*) and frequency (%), and measurable variables are defined using the arithmetic mean (average, avg.), standard deviation (SD), median (median), minimum (minimum), and maximum (maximum) values.

Due to the nature of the variables (measurable variables described on the ordinal scale and the lack of normality of the distribution of quantitative variables), non-parametric tests were used for statistical analyses.

The relationship between the dependent variable (fatigue level) and independent variables (demographic and clinical factors) was analyzed using the Mann–Whitney U test (this test concerns differences in the position of two unrelated samples) and the Kruskal–Wallis test (this test concerns differences in the place of at least three unrelated pieces). If the test probability p exceeded the assumed significance level pf α = 0.05, the distribution of variables in both groups was the same. However, post hoc Kruskal–Wallis multiple comparison tests were used to accurately check which groups had significant differences.

Correlations between measurable variables were checked using the significance test of Spearman’s rank correlation coefficient. If the test probability p exceeded the assumed significance level of α = 0.05, it meant no correlation existed between the examined variables.

Multiple linear regression and deepened multiple-step regression for the most significant coefficient were used to assess the influence of several independent variables (clinical factors) on the level of fatigue.

A *p*-value of <0.05 was considered statistically significant.

Statistical calculations were performed using the STATISTICA 10 PL statistical package.

## 3. Results

### 3.1. Characteristics of the Study Group

One hundred and twenty-eight patients with RA participated in the study. Women were the most numerous group (*n* = 110). The respondents were aged 19 to 83 years old. Eighty-eight people lived in the city; many people (*n* = 87) were married, while divorced individuals represented the smallest proportion (*n* = 4). Thirty-eight patients had higher education and 64 remained professionally active; only 5 were unemployed. Twenty-four patients were treated with biological drugs. The mean duration of the disease was 11 years, and the mean duration of morning stiffness was 53.3 min (Table 1 and Table 2). Over 58% of patients had at least one comorbid disease; approximately 37% suffered from hypertension, approximately 9% had type 2 diabetes, and approximately 7% had ischemic heart disease.

### 3.2. Measurement Scales

Table 3 shows the average fatigue value in the subjects on the FACIT-F scale, which was 24.1 ± 9.1 points, and on the SF-36 Vitality score scale, where the respondents obtained an average of 14.2 ± 1.8 points. These values indicate that people with RA experience chronic fatigue.

### 3.3. Relationships and Correlations between Clinical and Demographic Factors and the Level of Fatigue 

In the case of the following clinical and demographic factors, no statistical significance was demonstrated for the level of fatigue: Gender, age, education, marital status, place of residence, occupational status, body mass index (BMI), disease duration, pharmacological treatment, CRP, or RF.

There was a correlation between fatigue and pain (Rs = 0.370), where higher values of the VAS scale were accompanied by higher values of the FACIT-F scale, which meant that people experiencing more pain had more severe symptoms of fatigue. A positive correlation was also obtained between morning stiffness and fatigue (Rs = 0.217). People with longer-lasting morning stiffness have more severe fatigue symptoms because higher values accompany higher morning stiffness on the FACIT-F scale.

A negative correlation was observed between hemoglobin concentration and fatigue (Rs = −0.189) whereby people with normal hemoglobin levels have less severe symptoms of fatigue than people with lowered hemoglobin levels (lower FACIT-F values were accompanied by higher hemoglobin values). Positive correlations were also found between pressure soreness (Rs = 0.316) and disease activity (Rs = 0.258) and fatigue. People with more significant pressure soreness and disease activity had more severe symptoms of fatigue (higher values of the Ritchie Articular Index and DAS28 were accompanied by higher values of the FACIT-F scale) (Table 4). 

### 3.4. Multiple Regression

The results of multiple regression analysis for variable fatigue and three independent variable models are presented in Table 5.

Model 1 examined the combined effects of hemoglobin, CRP, and RF on fatigue. Hemoglobin was a significant predictor, explaining only 2.6% of the variation in the intensity of fatigue symptoms on the FACIT-F scale. Increasing the hemoglobin concentration by one unit reduced the power of fatigue symptoms on the FACIT-F scale by an average of 0.604 ± 0.286 points. 

In model 2, regression analysis for pharmacological treatment (biological/classical drugs) and disease activity (DAS28), with a significance level of *p* < 0.0045, showed that DAS28 is a significant factor for fatigue (FACIT-F). The regression model explained only 6.7% of the variation in the intensity of fatigue symptoms on the FACIT-F scale. Increasing the DAS28 disease activity by one point increased the power of fatigue symptoms in the FACIT-F scale by an average of 2.893 ± 0.877 points. 

Model 3 analyzed the effects of pain (VAS), morning stiffness, hemoglobin, joint tenderness (Ritchie Articular Index), and disease activity (DAS28) on fatigue. Significant factors for fatigue (FACIT-F) and significant variables were pain (VAS) and joint tenderness (Ritchie Articular Index). The regression model explained 18.0% of the variation in the intensity of fatigue symptoms on the FACIT-F scale. Increasing pain perception on the VAS scale by one centimeter increased the power of fatigue symptoms on the FACIT-F scale by an average of 1.258 ± 0.321 points. Increasing the Ritchie Articular Index by one point increased the severity of fatigue symptoms on the FACIT-F scale by an average of 0.094 ± 0.044 points. Pain sensation on the VAS scale had a more significant impact on the severity of fatigue symptoms on the FACIT-F scale than the Ritchie Articular Index.

## 4. Discussion

In this research, attempts were made to determine which factors, and to what extent, affect the level of fatigue, both negatively and positively. They were examined regarding demographic characteristics or psychosocial indicators in the clinical picture. Our research confirmed that patients with rheumatoid arthritis experience chronic fatigue to a significant degree. Many other scientists have also confirmed the presence of fatigue symptoms, such as the study by Druce et al. [14] using the SF-36 Vitality score, in which 38.8% of patients (*n* = 6835) reported severe fatigue. Hammam et al. [15], using the Multidimensional Assessment of Fatigue-Global Fatigue Index (MAF-GFI) questionnaire, showed that among Egyptian patients the mean values of fatigue were 27.2 ± 8.9 (*n* = 115), which indicates the presence of severe fatigue. Lao et al. [16] assessed the level of fatigue in elderly patients (≥60 years of age). They used the Multidimensional Fatigue Inventory-20 (MFI-20) and SF-36 Vitality scores to assess fatigue and showed that the elderly experience severe fatigue (MFI 60 ± 14), especially in the physical area. In the study by Abdel-Magied et al. [17], out of 50 examined patients, more than half (26 people) indicated a high level of fatigue (VAS > 50 mm and SF-36 Vitality scores—60.5 ± 23.1). Wagan et al. [18] determined the level of fatigue among Pakistani RA patients, in which 62% of participants (*n* = 192) showed a symptom of fatigue on the FACIT-F scale. Our research did not show the influence of demographic variables on fatigue severity. Uhlig and Provan [19], Feldthusen et al. [20], and Katz et al. [21] also indicated that factors such as age and gender were not significantly associated with fatigue. Different results can be seen in the studies by Lee et al. [22] and Diniz et al. [23]. These studies showed that younger age is significantly associated with higher fatigue values, but gender is not. In the study by Bączyk et al. [24], mean age was one of the predictors. van Steenbergen et al. [25], in a longitudinal study lasting eight years using the VAS scale (from 0 to 100 mm), proved that women had higher values than men (by 6.1 mm on the VAS scale). Moreover, Rodriguez-Muguruza et al. [26] and Olsen et al. [27] found that female sex and younger age influenced the level of fatigue. Sex, marital status, and social status did not show a statistically significant correlation with fatigue in Wagan et al., while age and education were correlated [18]. Relating these results to the results of the general population, it can be understood that the Swedish study’s aim was to describe the pattern of fatigue in the general population and explain the relationship between age and gender [28]. Their research showed that higher age significantly correlated with the level of fatigue, especially in the area of mental fatigue and general fatigue. Gender was also related to the level of fatigue; namely, women experienced significant fatigue on all subscales (Multidimensional Fatigue Inventory scale; MFI-20) compared to men [28].

Some clinical determinants correlated with fatigue. The influence of disease activity on the deterioration of fatigue results was also observed by Hammam et al. [15], Diniz et al. [23], Abdel-Magied et al. [17], Madsen et al. [29], Lao et al. [16], Bingham et al. [30], Holdren et al. [31], and Wagan et al. [18]. Different results, indicating no relationship between disease activity and the level of fatigue, were obtained by Feldthusen et al. [20], Lee et al. [22], Rodriguez-Muguruza et al. [26]. Madsen et al. [29], Lao et al. [16], Bingham et al. [30], Abdel-Magied et al. [17], Olsen et al. [27], Holdren et al. [31], and Bączyk and co-authors [24], confirming that increased pain and its changing nature increased fatigue levels. Lee et al. [22], Hammam et al. [15], and Wagan et al. [18] analyzed the influence of hemoglobin concentration on fatigue; their studies did not show a relationship between the variables, in contrast to the results of our studies. On the other hand, the influence of low hemoglobin values on fatigue was confirmed by Madsen et al. [29], van Steenbergen et al. [25], and Olsen et al. [27]. 

Our studies did not show any relationship between pharmacological therapy and an increase in fatigue. Ines et al. [32] conducted a cohort study where they assessed, among other things, the effect of biological treatment (anti-TNFα) on fatigue symptoms, and a significant decrease in the level of fatigue was noted. Janoudi et al. [33] confirmed that several months’ use of adalimumab reduces the symptoms of fatigue as measured by the FACIT-F scale. Similar results were obtained by Keystone et al. [34] and Michaud et al. [35], who analyzed the effect of baricitinib and adalimumab. Olsen et al. [27] investigated the impact of tumor necrosis factor (TNF-α) treatment in combination with methotrexate (MTX) and MTX alone. Patients continued to experience fatigue regardless of their disease activity status, even after a certain period of drug treatment. In their studies, Druce et al. [14] indicated that anti-TNFα therapy has positively affected the reduction of fatigue levels (SF-36 Vitality score).

The impact of morning stiffness on the severity of fatigue was confirmed by Abdel-Magied et al. [17] and Rodriguez-Muguruza et al. [26], as in our studies. Correlations between higher-pressure soreness in the joints and high values of fatigue were also obtained by Abdel-Magied et al. [17], Bingham et al. [30], van Steenbergen et al. [25], and Holdren et al. [31]. The results of studies by Madsen et al. [29] did not confirm that the number of affected joints and their tenderness increase the feeling of fatigue.

The available studies on the impact of various factors (clinical and demographic) on the intensity of fatigue show that determining which of them is a constant predictor of exacerbating the feeling of fatigue is difficult. It is essential that these determinants often do not affect the symptoms tested alone but collectively worsen the condition of patients in the area of fatigue. The regression analysis in this study confirmed the effect of pharmacological treatment and disease activity, where these predictors accounted for a total of 6.7% of the variability in the severity of fatigue symptoms. Another model, which turned out to be statistically significant, analyzed the combined assessment of the effects of pain, number of joints involved, hemoglobin level, morning stiffness, and disease activity on the level of fatigue. It turned out that only pain and the number of joints involved (i.e., how many joints) explain 18.0% of the variability in the severity of fatigue symptoms.

Corominas et al. [36] assessed the effect of tocilizumab on the level of fatigue and clinical factors affecting fatigue. Regression analysis showed that improvements in disease activity (DAS28), sleep, and depression explained 56% and 47% of the fatigue variance at 12 and 24 weeks of follow-up, respectively. In contrast, pain and hemoglobin levels have not been identified as significant factors. 

The pain component that influences the severity of fatigue symptoms was confirmed in their studies using linear regression by Madsen et al. [29], Oncü et al. [37], and Bouchaala et al. [38]. Gao et al. [39] showed that 59.5% of disease activity, self-efficacy, physical fitness, pain, depression, and the duration of morning stiffness and anxiety increased the sense of fatigue in RA patients. Van Steenbergen et al. [25] indicated in their analyses that each additional tender and painful joint and higher values of inflammation (CRP) increased the feeling of fatigue.

Feldthusen et al. [40] conducted multivariate analyses to determine monthly and seasonal changes in the severity of fatigue symptoms in Swedish RA patients. The winter period turned out to be the most critical factor contributing to the deterioration of the feeling of tiredness in physical and mental terms. Hammam et al. [15] indicated that the variables sleep, depression, and disease activity explain 49.1% of the variability in fatigue in a multivariate analysis for RA patients. Sleep quality was the essential factor in determining fatigue.

Rinke et al. [41] indicated that in the multiple analysis of multiple factors, female gender, physical well-being, and the Rheumatoid Arthritis Impact of Disease (RAID) were significant predictors of fatigue in patients starting treatment with biological drugs. Ultimately, female gender still resulted in being the parameter that most strongly influenced fatigue. Pain, physical well-being, and disease activity were negligible factors in changing the level of fatigue. 

Ifesmen et al. [42] studied the influence of various factors on the intensity of fatigue in RA patients for three years in a group of 1236 patients (at the beginning of the study, after six months, and then every year). Of the determinants tested, only female gender, poor mental health, pain, higher Health Assessment Questionnaire, and the number of tender joints were significantly associated with vitality (as measured by the SF-36 Vitality score) in a multiple regression analysis. Together, these factors accounted for approximately half of the variability in fatigue at baseline (R^2^ = 0.49), where mental health and pain were the factors that most influenced the intensity of the fatigue level.

Based on the results of statistical research (correlation and multiple regression), it can be concluded from the analyzed parameters that pain and disease activity had the greatest impact on the development of fatigue. However, the study’s authors would not be so radical in determining one or two main factors because they believe that fatigue is influenced by several parameters simultaneously.

Limitations of the study, such as the small number of patients, the lack of research on the reproducibility of the results in the same subjects (especially patients in the period of active disease and then remission or after changing pharmacological treatment), or failure to take into account other factors (e.g., comorbidities, mental condition) open the field for conducting further in-depth analyses of the impact of fatigue on patients with RA and allowing for an understanding of the existing problems and establishing a strategy/concept for coping with it.

The innovation of this study is that a large number of clinical and demographic variables were analyzed to determine which factors influence the level of fatigue and how (increasing its intensity or reducing it). Based on the performed progressive multiple-step regression, it has been shown that it is possible to theoretically predict which parameters (their intensity or reduction) affect the intensity of the fatigue level.

## 5. Conclusions

Our research has shown that patients with RA present symptoms of chronic fatigue to a significant degree. The relationship between demographic factors and certain clinical factors with the level of fatigue has not been demonstrated. On the other hand, the level of fatigue was correlated with pain, long-lasting morning stiffness, the active form of the disease, increased soreness of the joints, and low hemoglobin values.

Based on progressive multiple-step regression, it has been shown that it is possible to theoretically predict which parameters (and their severity or reduction) affect the intensity of fatigue levels. Our study showed that hemoglobin (increasing its concentration reduced the severity of fatigue symptoms) and pain, disease activity, and tenderness (increasing their perception will increase the severity of fatigue) impacted fatigue levels.

We believe that fatigue symptoms should be considered in every interview with patients, both by nursing and medical staff. When analyzing the symptoms of fatigue, each patient should be approached individually, using the existing questionnaires or asking key questions to recognize the situation. The presence of fatigue symptoms should be considered during therapy and patient care by searching for and eliminating additional, intensifying stimuli and increasing its level.

## Figures and Tables

**Table 1 ijerph-19-14681-t001:** General characteristics of the respondents for demographic variables.

Demographic Variables	Female *n* = 110 (85.9%)		Male *n* = 18 (14.1%)		Total *n* = 128	
	*n*	%	*n*	%	*n*	%
Place of residence						
City	77	70.0	11	61.1	88	68.8
Country	33	30.0	7	38.9	40	31.3
Marital status						
Single	21	19.1	1	5.6	22	17.2
Married	72	65.5	15	83.3	87	68.0
Divorced	4	3.6	0	0.0	4	3.1
Widowed	13	11.8	2	11.1	15	11.7
Education						
Elementary school education	4	3.6	1	5.6	5	3.9
High school education	73	66.4	12	66.7	85	66.4
Higher education	33	30.0	5	27.8	38	29.7
Occupational status						
Professionally active	53	48.2	11	61.1	64	50.0
Unemployed	5	4.5	0	0.0	5	3.9
Retired	35	31.8	5	27.8	40	31.3
Pensioner	17	15.5	2	11.1	19	14.8
Residence status						
With someone	96	87.3	17	94.4	113	88.3
Alone	14	12.7	1	5.6	15	11.7
Biological treatment						
Yes	19	17.3	5	27.8	24	18.8
No	91	82.7	13	72.2	104	81.3
Age [years]						
Avg. ± SD	53.7 ± 14.4		54.6 ± 15.6		53.8 ± 14.5	
Median (min.–max.)	56.0 (19–83)		57.5 (23–75)		56.0 (19–83)	

Abbreviations: Avg.—average; SD—standard deviation; min.—minimum value; max.—maximum value.

**Table 2 ijerph-19-14681-t002:** General characteristics of the participants for clinical variables.

Clinical Variables	Female *n* = 110 (85.9%)	Male *n* = 18 (14.1%)	Total *n* = 128
Disease duration [years]			
Avg. ± SD	11.0 ± 9.0	11.5 ± 6.4	11.1 ± 8.7
Median (min.–max.)	8.5 (0.2–40)	12.5 (1–22)	9.5 (0.2–40)
Visual Analogue Scale for pain (VAS) [cm]			
Avg. ± SD	5.7 ± 2.4	5.2 ± 2.6	5.6 ± 2.4
Median (min.–max.)	6.0 (0–10)	6.0 (0–9)	6.0 (0–10)
Morning stiffness [min]			
Avg. ± SD	54.8 ± 72.6	43.8 ± 42.2	53.3 ± 69.1
Median (min.–max.)	30.0 (0–420)	30.0 (0–120)	30.0 (0–420)
Hemoglobin (Hgb) [g/dL]			
Avg. ± SD	9.2 ± 2.5	9.7 ± 4.1	9.3 ± 2.8
Median (min.–max.)	8.3 (3.4–14.2)	9.0 (5.0–23.2)	8.4 (3.4–23.2)
C-reactive protein (CRP) [mg/L]			
Avg. ± SD	12.3 ± 16.4	6.8 ± 8.2	11.5 ± 15.6
Median (min.–max.)	6.0 (0.1–84.4)	4.3 (0.3–32.8)	5.9 (0.1–84.4)
Rheumatoid factor (RF) [IU/mL]			
Avg. ± SD	74.7 ± 127.0	103.1 ± 167.6	78.7 ± 133.1
Median (min.–max.)	30.2 (1–650)	36.0 (10–650)	32.8 (1–650)
Ritchie Articular Index [pts]			
Avg. ± SD	23.6 ± 17.2	27.3 ± 20.9	24.2 ± 17.7
Median (min.–max.)	21.5 (1–53)	23.0 (3–53)	21.5 (1–53)
Disease Activity Score 28 (DAS28)			
Avg. ± SD	3.8 ± 0.9	3.9 ± 1.0	3.8 ± 0.9
Median (min.–max.)	4.0 (1.8–5.9)	4.0 (2.3–5.1)	4.0 (1.8–5.9)

Abbreviations: Avg.—average; SD—standard deviation; min.—minimum value; max.—maximum value.

**Table 3 ijerph-19-14681-t003:** Fatigue on the FACIT-F scale and vitality on the SF-36 scale vitality scores.

Scales	Avg. ± SD	Median	Min.	Max.
FACIT-F [pts]	24.1 ± 9.1	24.0	7	49
SF-36 v.s. [pts]	14.2 ± 1.8	14.0	8	20

Abbreviations: FACIT-F scale—a maximum of 52 points can be obtained indicating very strong fatigue and a minimum of 0 points indicating no fatigue. The range of points in the question is 0–4. Scale SF-36 v.s.—a maximum of 24 points can be obtained indicating high quality of life and a minimum of 4 points indicating low quality of life. The range of points in questions 1–6.

**Table 4 ijerph-19-14681-t004:** The results of the significance test of Spearman’s rank correlation coefficient between the independent variables and the FACIT-F scale.

Variables	FACIT-F		
	Rs	*t* (n − 2)	*p*
Visual Analogue Scale for pain VAS [cm]	0.370	4.47	0.0000
Morning stiffness [minutes]	0.217	2.49	0.0140
Hemoglobin (Hgb) [g/dL]	−0.189	−2.16	0.0325
Ritchie Articular Index [pts]	0.316	3.74	0.0003
Disease Activity Score 28 (DAS28)	0.258	3.00	0.0032

Abbreviations: Rs—the value of the Spearman coefficient for the number n, *t*—the value of the *t* statistic checking the significance of the Rs coefficient for the number of degrees of freedom n − 2, *p*—probability level *p* for the *t* statistic.

**Table 5 ijerph-19-14681-t005:** Multiple regression results for the FACIT-F variable and independent variables.

Variables	b *	Std. Error for b *	b	Std. Error for b	*p*
Model 1 R = 0.195; R^2^ = 0.038; Corrected R^2^ = 0.015; F (3.124) = 1.63; *p* < 0.1858; Std error estimation: 8.99 *					
Hemoglobin (Hgb) [g/dL]	−0.181	0.091	−0.591	0.296	0.0479
C-reactive protein (CRP) [mg/L]	−0.003	0.097	−0.002	0.056	0.9762
Rheumatoid factor (RF) [IU/mL]	0.062	0.094	0.004	0.006	0.5136
R = 0.185; R^2^ = 0.034; Corrected R^2^ = 0.026; F (1.126) = 4.47; *p* < 0.0365; Std error estimation: 8.93 **					
Hemoglobin (Hgb) [g/dL]	−0.185	0.088	−0.604	0.286	0.0365
Model 2 R = 0.285; R^2^ = 0.081; Corrected R^2^ = 0.067; F (2.125) = 5.53; *p* < 0.0045; Std error estimation: 8.75 *					
Pharmacological treatment	0.046	0.089	1.070	2.066	0.6053
Disease Activity Score 28 DAS28	0.295	0.089	2.893	0.877	0.0013
R = 0.282; R^2^ = 0.079; Corrected R^2^ = 0.072; F (1.126) = 10.86; *p* < 0.0013; Std error estimation: 8.72 **					
Disease Activity Score 28 (DAS28)	0.282	0.085	2.764	0.839	0.0013
Model 3 R = 0.440; R^2^ = 0.193; Corrected R^2^ = 0.180; F (2.125) = 14.97; *p* < 0.0000; Std error estimation: 8.20 **					
Visual Analogue Scale for pain VAS [cm]	0.338	0.086	1.258	0.321	0.0001
Ritchie Articular Index [pts]	0.184	0.086	0.094	0.044	0.0349

* multiple regression; ** progressive multiple-step regression; R—correlation coefficient; R^2^—correlation of determination; F—F test statistic; b—regression coefficient.

## Data Availability

The data are not publicly available due to data privacy regulations. The data presented in this study are available upon request from the corresponding author.

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
