# Peer review of "The Link between the Demographic and Clinical Factors and Fatigue Symptoms among Rheumatoid Arthritis Patients"

_ijerph, 2022, doi:10.3390/ijerph192214681_

Round 1

Reviewer 1 Report

The manuscript titled: The link between the demographic and clinical factors and fatigue symptoms among rheumatoid arthritis patients (ijerph-2010395) is interesting. It presents the relationship between fatigue and demographic and clinical aspects in 128 RA patients. The author found a connection between the level of fatigue and pain, long-lasting morning stiffness, active disease, increased soreness of joints, and low hemoglobin values. The manuscript is well-written; however few things require explanation:

1.       What kind of EURAR criteria was used to diagnose RA? There is no reference.

2.       How were the patients recruited? Are these patients recruited from outpatient clinics or hospital departments?

3.       Who measured Ritchie Articular Index and disease activity DAS28? How was this procedure conducted? What kind of DAS28 did you calculate? There are various procedures of DAS28 estimation used in clinical practice.

4.       Did the analyzed patients have internal organ involvement? Authors in the introduction describe the possible organ involvement in the course of RA (heart, lungs, kidneys, or eyes), but they are not described in the study group. 

5.       The mean age of the study group was about 54 years. Is there available data that describes the general population's fatigue level in the age-gender-matched groups?

6.       The authors concluded that the level of fatigue correlated with pain, long-lasting morning stiffness, the active form of the disease, increased soreness of the joints, and low hemoglobin values. Which of the analyzed parameter can have the strongest influence on fatigue development?

If the authors consider the above comments, I believe the work is valuable and can be published in the International Journal of Environmental Medicine and Public Health. 

Author Response

We would like to thank the Reviewers for their valuable comments. We included them all in the revised version of the article. We trust that thanks to them, our article has become more interesting, will be appreciated by the Editors, and will be accepted for publication.

In the manuscript, corrections and additions are marked in red.

Reviewer 2 Report

·         No plagiarism detected – evaluated using Ithenticate (report attached).

·      According to instructions for authors: “When defined for the first time, the acronym/abbreviation/initialism should be added in parentheses after the written-out form”. Ex1: some acronyms are used in abstract without previous written out form; Ex2: in line 42 the acronym is used: “OMERACT (Outcome Measures in Rheumatology)”, when it should be Outcome Measures in Rheumatology (OMERACT); EX3 (line 104): ESR and VAS not defined before.

·         Rheumatoid arthritis can be replaced by “RA” in the article after definition is given.

··         While the written presentation is mostly acceptable, there are a number of problematic statements throughout due to use of non-precise verbiage. These must be corrected through careful editing or engagement of native English speaker or editorial service. Ex1 (lines 21-22) “Changing the values of parameters such as pain, disease activity, tenderness, and hemoglobin may reduce or worsen the feeling of fatigue”; Ex2 (lines 39-40) “In addition to the symptoms listed above, fatigue is a familiar feeling reported by RA patients. It appears in over 70 percent of the sick.”; Ex3 (line 74) “and who was utterly logical”.

·         Some statements lack reference.

·         Exclusion criteria of enrolled persons is not adequately presented.

·        Table 1 can be improved so that the Demographic variables are aligned with the corresponding values, and so it fits into a single page. Decimals are used with (,) and (.). Please keep consistent.

·         Line 350 “.t”?

·         Paragraph lines 327-333: different font than the rest of the document.

·       Please be sure to follow reference guidelines. According to the Reference List and Citations Style Guide for MDPI Journals: “For documents co-authored by a large number of persons (more than 10 authors), you can either cite all authors, or cite the first ten authors, then add a semicolon and add ‘et al.’ at the end: Author 1; Author 2; Author 3; Author 4; Author 5; Author 6; Author 7; Author 8; Author 9; Author 10; et al.” Please adjust accordingly.

Author Response

(The authors gave the same response as above.)

Round 2

Reviewer 2 Report

Accept in current form